# Immunological and Neurological Signatures of the Co-Infection of HIV and HTLV: Current Insights and Future Perspectives

**DOI:** 10.3390/v17040545

**Published:** 2025-04-08

**Authors:** Md. Nazmul Islam, Masuma Akter Mili, Israt Jahan, Cotton Chakma, Rina Munalisa

**Affiliations:** 1Department of Neuroscience of Disease, Brain Research Institute, Niigata University, 1-757, Asahimachidori, Chuo-ku, Niigata 951-8585, Japan; 2Department of Microbiology, Noakhali Science and Technology University, Noakhali 3814, Bangladesh; masumamili3@gmail.com (M.A.M.); isratjahanrozy214@gmail.com (I.J.); cottonchakma5053@gmail.com (C.C.); 108727110@gms.tcu.edu.tw (R.M.)

**Keywords:** HTLV, HIV, co-infection, immune response, neurological disorder

## Abstract

The human retroviruses HIV and HTLV-1/HTLV-2 are transmitted through similar pathways but result in markedly different diseases. This review delineates the immune-mediated mechanisms by which HTLVs influence HIV pathology in co-infected individuals. In the context of HIV co-infection, HTLV-1/HTLV-2 alter the cellular microenvironment to enhance their own survival while simultaneously impeding the progression of HIV. Despite the extensive body of literature on the biological and clinical implications of retroviral co-infections, decades of research have been marred by controversy due to several flawed epidemiological studies and anecdotal reports lacking robust statistical and scientific backing. Nevertheless, recent systematic and well-designed research has led to a growing consensus supporting at least three key conclusions: (1) co-infections of HIV-1 and HTLV-1 are frequently observed in patients with elevated CD4^+^ T-cell counts who present with lymphoma or neurological complications; (2) HIV-1 and HTLV-2 co-infections have been associated in some instances with a “long-term non-progressor” phenotype; (3) the differential function and/or overexpression of the HTLV-1 and HTLV-2 Tax proteins are likely crucial in the clinical and immunologic outcomes of HIV/HTLV-1 and -2 co-infections. The present review will provide a comprehensive account of research on retroviral co-infections, focusing particularly on their clinical manifestations and associated pathological features.

## 1. Introduction

Co-infections of HIV with human T lymphotropic virus type 1 (HTLV-1) and type 2 (HTLV-2) are common in metropolitan areas where injection drug use is prevalent. Previous studies have indicated that individuals co-infected with HIV and HTLV-1 exhibit elevated CD4 cell counts, while those co-infected with HIV and HTLV-2 show elevated CD8 cell counts [1]. Although HTLV has been infecting humans for millennia, understanding of its infection and pathogenesis has only recently begun to emerge. The virus can be transmitted from mother to child, through sexual contact, and via contaminated blood products. In regions such as Japan, sub-Saharan Africa, the Caribbean, and South America, over 1% of the general population is infected with HTLV-1. While most HTLV-1 carriers remain asymptomatic, the virus is linked to severe diseases [2]. It is estimated that 10–20 million people globally are infected with HTLV-1. HIV and HTLV-1, both of which are classified as retroviruses, have been observed to target CD4^+^ T cells. The resulting effects on the nervous system have been demonstrated to manifest in a variety of neurological disorders. For example, Asymptomatic Neurocognitive Impairment (ANI) and Mild Neurocognitive Disorder (MND), which may result from HIV infection, can progress to HAND. Although the majority of these individuals remain asymptomatic, the virus is associated with particularly severe conditions, including adult T-cell leukemia/lymphoma (ATL) and an inflammatory disease of the central nervous system known as HAM/TSP [3].

The most significant complications of HTLV-1 infection include HAM, also referred to as TSP, and ATL, each carrying a lifetime risk of 3–5%. Recent research has expanded the focus to encompass a broader spectrum of potential central nervous system (CNS) complications linked to HTLV-1, alongside heightened efforts toward prevention and enhanced treatment strategies for HTLV-1-related diseases. [4]. HTLV-1 establishes a chronic infection in T lymphocytes, predominantly persisting in a proviral state. While most infected individuals remain asymptomatic, the host’s T-cell immune response plays a critical role in determining the likelihood of secondary disease development. HTLV-1 indirectly induces CNS damage by infecting CD4^+^ T cells, which migrate across the blood–brain barrier and trigger the activation of cytotoxic CD8^+^ T cells, ultimately resulting in neuroglial cell death and degeneration [5]. The risk of developing HAM/TSP is influenced by both the duration of infection and the proviral load. Similar to other HTLV-1-associated conditions, the precise mechanism underlying HAM/TSP pathogenesis remains incompletely understood; however, it is thought to involve an immune response to viral antigens, leading to T-cell activation [6]. At present, no curative treatment exists for HAM/TSP, but therapeutic interventions can alleviate symptoms and slow disease progression

In this review, we highlight the unique clinical features and diagnostic challenges encountered in managing the co-infection of HIV and HTLV. This study offers a multidisciplinary, in-depth analysis of the epidemiology, clinical manifestations, and therapeutic approaches for HAM/TSP, drawing on a comprehensive review of the current literature. The objective is to advance understanding of this rare and intricate disorder and to underscore the critical need for early diagnosis and effective management of its clinical features.

## 2. Effects of Co-Infection on Immune System Dynamics

HTLV-1 and HIV both predominantly infect CD4^+^ T cells and have comparable modes of transmission [7]. Detection of HTLV/HIV co-infection in patients solely infected with HIV is not consistently feasible; nonetheless, HIV patients with co-infections generally exhibit elevated CD4^+^ T-cell counts compared to those with mono-infection who progress to AIDS. Furthermore, HTLV/HIV co-infection is associated with a heightened prevalence of activated CD4^+^ T cells expressing CD45RO and CD25 [8]. Both HIV and HTLV-I are retroviruses that target CD4^+^ T cells, leading to the disruption of their normal functions [9]. Although patients who are co-infected with HIV and HTLV-1 have been shown to have higher CD4^+^ T-cell counts, the lymphoproliferative effect of HTLV-1 may create a misleading sense of immunological competence. This could result in the postponement of antiretroviral therapy (ART) based on CD4^+^ count guidelines [10]. Tax protein typically exerts its effects on HIV expression directly, and in the extracellular compartment, it has been observed to stimulate the immune system. However, this protein has been demonstrated to enhance the replication of HIV-1 through the stimulation of the HIV-1 long terminal repeat (LTR). The consequence of HIV-1/HTLV-1 co-infection is the overproduction of lymphocytes that are defective in their functioning, in response to the elevated levels of HIV-1 virion production in the environment [11]. Additionally, Tax-1 has been found to increase the expression of various cellular proteins, notably, transcription factors and cytokines like IL-2 and TNF-α (Figure 1). Research has shown that certain cytokines, including TNF-α and IL-1β, can enhance HIV-1 transcription via an NF-κB-dependent pathway [12].

The presence of HIV co-infection is a significant factor influencing the systemic cytokine profile of individuals with dual infections. In contrast to the immunosuppression observed in immunodeficiency, these patients exhibit a proinflammatory environment conducive to chronic inflammatory conditions, which can be further exacerbated in certain instances [13]. In individuals co-infected with HIV and HTLV-1, a higher concentration of Th1 cytokines (IL-2 and IFNγ) were observed in peripheral blood mononuclear cells (PBMC) when compared to PBMC from individuals infected with either HIV or HTLV-1 alone, suggesting that HTLV-1 plays a dominant role in the pathogenesis of this condition, potentially at the expense of HIV-1 infection [14]. Individuals infected with both HIV and HTLV-1 show elevated levels of proinflammatory cytokines, including interleukin 2 and interferon-γ. This suggests that chronic immune activation may contribute to the development of more severe metabolic and cardiovascular complications [14].

Co-infection with HTLV-1 and HIV, common in tropical areas, disturbs immunological dynamics by fostering chronic inflammation, impaired cell proliferation, and increased viral replication. This interaction can obscure immunodeficiency, complicating treatment for HIV and increasing the risk of severe inflammatory and metabolic consequences.

## 3. Viral Interplay and Synergistic Pathogenesis: Mechanisms of Co-Infection Between HIV and HTLV-1

Microbes that affect the same host may engage in either competitive exclusion or mutualistic interactions, whereby the presence of one microbe positively influences the replication of another. Recent findings suggest that a microorganism may enhance its own reproduction by modulating the host immune system, thereby either attenuating or facilitating the infection of other pathogens [15]. Numerous epidemiological investigations have shown that HTLV-1 infection intensifies the cytopathic effects of HIV-1 and hastens the evolution of AIDS symptoms in co-infected persons [14]. The progression of HIV-1 disease was more accelerated and the mean survival time was shorter in co-infected individuals than in those infected with HIV-1 alone, in the context of HTLV-1/HIV-1 co-infection [8]. It has been demonstrated that co-infection with HIV and HTLV-I may result in an increase in the HIV viral load, while simultaneously leading to a reduction in the HTLV-I proviral burden [16].

Host genetic factors play an important role in this variability and the pathogenesis of HIV and HTLV co-infection. For example, the CCR5 (chemokine (C-C motif) receptor 5) co-receptor is necessary for infection with R5 strains of HIV, and it also observed that individuals who are homozygous for the CCR5Δ32 deletion mutation represent the only known genotype capable of protecting against HIV infection [17].

Retroviral infections, such as HIV and HTLV-1, cause neuronal damage that leads to structural and functional changes in the brain that affect a wide range of functions [18]. A recent study indicated a 10.9% prevalence of HIV/HTLV co-infection among HIV-positive individuals from an HTLV-endemic region associated with an increased risk of neurological illnesses. Furthermore, it was revealed that ART did not confer protection against these conditions nor alter the proviral HTLV load or CD4^+^ lymphocyte count [19].

## 4. Longitudinal Patterns in HIV and HTLV-1 Co-Infection: Viral Loads, Immune Responses, and Disease Trajectories

One of the earliest studies in this field, conducted in 1984, demonstrated that approximately 7% of patients with AIDS in Haiti were also infected with HTLV-1 [20]. The majority of research on HTLV-1 and HIV-1 co-infection is currently concentrated in South America and Africa [19]. In certain areas, such as New Orleans in Louisiana, up to 5% of individuals infected with HIV may also be infected with HTLV-1 or HTLV-2 [21].

A comparison of individuals infected with HIV-1 alone and those coinfected with HTLV-1 revealed that the latter group typically exhibits more advanced stages of HIV-1 infection [22]. The CD4^+^ cell count was found to be higher in patients co-infected with HIV-1 and HTLV-1 compared to those infected with HIV-1 alone. This is due to the fact that HTLV-1 has been observed to stimulate the growth of CD4^+^ lymphocytes [23]. Furthermore, additional research has demonstrated that individuals co-infected with HIV-1 and HTLV-1 exhibit elevated CD4^+^ and CD8^+^ cell counts relative to those infected with a single virus [23]. Despite the presence of sufficient levels of CD4^+^ cells, HIV-1/HTLV-1 co-infection has been demonstrated to be significantly associated with a shorter survival time [24].

Consequently, in the event of co-infection, monitoring ART response using the CD4^+^ T-cell count as a measure may prove to be an ineffective strategy [25,26]. The expansion of T cells in response to HTLV-1 infection is typically counterbalanced by a higher rate of infected cell death as a consequence of immune surveillance [27]. In individuals with HIV-1/HTLV-1 co-infections, there is an increase in the number of lymphocytes with impaired functionality. This is due to the fact that HTLV-1 stimulates the production of T cells, resulting in the generation of a substantial number of poorly functioning lymphocytes in the context of HIV coinfection. This leads to an environment that is conducive to the production of high levels of HIV-1 virions [28]. HIV/HTLV-1 co-infections also have a strong link with tuberculosis (TB), which is caused by *Mycobacterium tuberculosis*. Individuals infected with HIV are at an elevated risk of developing TB compared to those who are HIV-negative. In resource-limited settings, TB is the leading cause of mortality among HIV-positive adults [29]. The precise nature of the relationship between HTLV-1 and TB remains uncertain [30]. A study conducted in Guinea-Bissau revealed that HTLV-1 infection was linked to the development of TB in HIV-positive patients, but not in HIV-negative patients. Another study proposed that HTLV-1 infection may be associated with elevated mortality rates among TB patients; however, a statistical analysis was not conducted [31].

## 5. Antiretroviral Therapy in HIV/HTLV-1 Co-Infection: Mechanisms, Efficacy, and Side Effects

Antiretroviral treatment and a vaccination specific to HTLV-1 have not yet been produced [32,33]. In individuals co-infected with HTLV-1, the HTLV-1 proviral load rises in response to antiretroviral therapy (ART) consisting of zidovudine, lamivudine, and abacavir (or didanosine), which was first administered to treat HIV-1 infections in patients with co-infections of HIV-1 and HTLV-1 [34]. The five distinct stages of the HIV life cycle that antiretroviral medications target consist of binding, fusion, reverse transcription, integration, and proteolytic cleavage. Commonly prescribed medications in resource-rich areas often prevent enfuvirtide fusion and CCR5 binding. HIV integrate aids DNA integration, reverse transcriptase catalyzes transcription, and protease cleaves polypeptide chains to prevent the last stage of maturation [35].

When HIV patients receive antiretroviral medication, their immune system is strengthened, viral replication is suppressed, opportunistic infections are reduced, and their lifetime is increased to almost normal. Common side effects of antiviral medication include dyslipidemia, altered glucose metabolism, hypersensitivity responses, hyperlactatemia/lactic acidosis, changes in body composition, and cardiovascular disease [36].

## 6. Therapeutic Efficacy of Antiretroviral Regimens in HIV and HTLV-1 Co-Infected Patients

Most HIV-1/HTLV-1 co-infected individuals are more likely to develop myelopathy, bronchitis, thrombocytopenia, urinary tract infections, or opportunistic diseases than HIV-1 mono-infected individuals, regardless of ethnicity, age, or CD4^+^ T-cell count [37]. Co-infected individuals had a higher incidence of neurological complications, including myelopathy (HAM/TSP) associated with HIV-1/HTLV-1 and peripheral neuropathy (PN) associated with both HIV-1/HTLV-1 and HIV-1/HTLV-2 co-infection [38].

HIV and HTLV-1 co-infection increases mortality despite higher baseline CD4^+^ counts. Although ART use decreases mortality, using CD4^+^ cell counts to guide ART initiation and prophylaxis timing may delay effective treatment and increase mortality [39]. Antiretroviral therapy with lamivudine, zidovudine, and abacavir (or didanosine) for HIV-1 infection in patients co-infected with HIV-1/HTLV-1 leads to an elevation in HTLV-1 proviral load [40]. An elevated occurrence of HTLV-1-associated neurological illness may be noted in co-infected persons undergoing antiretroviral medication, as ART effectively addresses HIV-1 but has a diminished impact on HTLV-1 expression [38]. Despite the existence of established protocols for the treatment of HIV-1 and HTLV-1 mono-infection, the dearth of well-documented therapeutic options for HIV-1/HTLV-1 co-infection may result in suboptimal or even adverse outcomes, underscoring the necessity of dual testing and customized treatment regimens. [10].

Individuals infected with both viruses exhibit an elevated prevalence of neurological disorders. The two most significant neurological consequences of HIV-1/HTLV-1 and HIV-1/HTLV-2 co-infection are myelopathy (related to HIV-1/HTLV-1) and PN (associated with either HIV-1/HTLV-1 or HIV-1/HTLV-2). The currently available antiretroviral drugs have not been demonstrated to be beneficial for individuals infected with HTLV-1/2 alone or with other viruses concurrently [38]. Despite elevated baseline CD4^+^ counts, co-infection with HIV-1 and HTLV-1 has been demonstrated to exacerbate a number of complications, including neurological complications, opportunistic infections, and mortality rates. Although ART has been shown to diminish HIV-related mortality, its limited impact on HTLV-1 expression, as well as the absence of standardized protocols for treating individuals with dual infections, emphasize the need for more personalized therapeutic strategies to enhance outcomes.

## 7. HIV-1 and HTLV-1 Reservoirs: Mechanisms of Persistence, Immune Evasion, and Implications for Treatment

Due to a tiny pool of infected CD4^+^ T cells, primate immunodeficiency viruses such as HIV-1 may infect a variety of cell types, live in lymphoid tissues, blood, and immune-privileged areas, and remain in a reservoir for an extended period of time even in patients receiving highly active antiretroviral treatment. Transitional memory cells (TCM) are preferred for persistence due to proviral DNA detection, whereas early viral reservoirs, primarily CD4^+^ T cells, impede sterilizing immunity, antiretroviral medications, and persist in central memory TCM cells, facilitated by T-cell survival and low-level antigen-driven proliferation [41,42,43]. A subset of cells known to be more susceptible to HIV infection is the memory CD4^+^ T cell reservoir, which is made up of transcriptionally silent DNA proviruses within these cells. These subsets include CD4^+^ T cells that express exhaustion markers like cytotoxic lymphocyte-associated protein 4 (CTLA-4) and central and stem cell memory CD4^+^ T cells, which survive through low-level cycling or self-renewal without activation [41].

According to some studies, genomic integration sites that are transcriptionally quiet or frequently transcribed are preferred by complete HIV-1 proviruses. This finding raises the possibility that the HIV-1 reservoir is similar to that of HTLV-1 [44]. The in vivo persistence mechanism of HIV-1 is notably different from that of HTLV-1. Whereas HIV-1’s cytolytic expression generally entails persistent de novo infection, HTLV-1’s noncytolytic expression permits clones to survive on occasion. During very vigorous ART, the reservoir of HIV-1-infected cells can last forever [45]. By reproducing predominantly through the clonal proliferation of infected cells, HTLV-1 minimizes the requirement for viral antigen production and immune-mediated death and expresses the proviral plus strand in infrequent, self-limiting bursts, and it maintains a balance between host immunological response and viral replication [45]. RNA reverse transcription by related RTs is necessary for HIV-1 and HTLV-1 retrovirus replication, indicating that anti-HIV-1 medications that target HIV-1 RT may also be effective against HTLV-1 RT. However, while antiretroviral medication treatment may effectively control HIV-1 infection and the associated disorders in most cases, there is currently no effective treatment for HTLV-1-caused inflammatory or malignant diseases [46]. Breastfeeding protects newborns from HTLV-1-infected mothers, suggesting that neutralizing antibodies can stop infection. Researchers have discovered that a particular polypeptide encoded in an anti-HTLV-1 vaccination has been shown to be both safe and effective in mice. TNF-α, IL-2, perforin, IFN-γ, and autologous CD8^+^ T cells were activated by ATLL patient monocytes [34].

## 8. Comparative Impact on Morbidity and Mortality in Co-Infected Versus Mono-Infected

Higher rates of HTLV-1/HIV-1 co-infection are reported in South America, the Caribbean and Africa. Research suggests that co-infection with HIV-1 and HTLV-1 may alter the natural history of HIV-1, resulting in faster clinical progression to AIDS and reduced survival. In a study conducted in 2001 by Brites and colleagues [8], it was observed that individuals infected with both HIV-1 and HTLV-1 exhibited a shorter average lifespan compared to those infected with HIV-1 alone, irrespective of gender or CD4 cell count. Additionally, Sobesky et al. observed that individuals co-infected with HIV-1 and HTLV-1 in French Guiana exhibited a heightened risk of mortality compared to those infected with HIV-1 alone. However, the limited sample size of their study may have constrained the strength of their conclusion [24]. HAM has been observed in 2–3% of cases [47,48] with HTLV-1 mono-infection, whereas three out of four co-infected individuals in an HIV cohort developed HAM. The hypothesis that HTLV-1 co-infection increases the prevalence of myelopathy is substantiated by subsequent prospective case-control studies that have compared individuals with HIV mono-infection to those who are co-infected [47,49].

The majority of HIV-1/HTLV-1 co-infected individuals, regardless of age, race, or CD4^+^ T cell count, are more likely to experience a range of complications, including myelopathy, thrombocytopenia, bronchitis, urinary tract infection, or opportunistic illness, compared to those infected with HIV-1 alone [37].

## 9. Neurological Effect

### 9.1. Neurological Complications in Patients Co-Infected with HIV and HTLV-1

Because the HIV and HTLV viruses have similar modes of transmission and are widespread in many communities, they frequently co-infect. Via both direct and indirect methods, these viruses induce severe immunological dysfunction and neuropathogenic neurological disorders. Although HTLV-1 produces the “HTLV-1 neurological complex” of immune-mediated disorders, of which HAM/TSP is the most prevalent manifestation, HIV-1 causes neurological disease either directly or indirectly [38,50]. Some research has indicated that the neurological manifestations observed in patients with HTLV-1 may be attributable to the localization site of the virus. This finding suggests a potential link between the involvement of the brain, cerebellum, and cranial nerves, and the subsequent development of neurological symptoms. Conversely, compromise of anterior horn cells, peripheral nerves, and muscles may be observed [51].

HIV-1/HTLV-1 or HIV-1/HTLV-2 co-infection can cause neurological symptoms such as myelopathy (HAM/TSP) and PN, which are more common in coinfected persons with HIV. HIV frequently causes PN, vacuolar myelopathy, neurocognitive problems, opportunistic infections, and CD8^+^ T-cell encephalitis. Since the advent of ART, PN has emerged as the most prevalent neurological consequence in HIV-positive individuals, affecting between 30% and 67% of those who are infected [10,38,52]. HIV-1/HTLV-1 co-infections have greater incidences of neurological conditions, including PN and myelopathy. Patients who are co-infected or have HTLV-1/2 infection alone have not demonstrated any benefits from the current antiretroviral treatments.

### 9.2. HIV-Associated Neurocognitive Disorders (HAND): Pathogenesis, Impact, and the Role of the CNS

Before ART, 20–30% of infected individuals had HIV-associated dementia. Other neurocognitive symptoms such as HAND are still common even though ART has decreased the occurrence of dementia to less than 5%. This disorder, which is more prevalent in those with mild cognitive impairments, can affect daily functioning and drug compliance. HAND is still a major problem in HIV-positive people even after ART [38]. It has been observed that the CNS functions as a significant reservoir for HIV. For instance, microglia and astrocytes within the CNS have been shown to be susceptible to HIV-1 and the NLRP3 inflammasome, which have been implicated in the development of neurological disorders (Figure 2).

A range of neurocognitive dysfunctions associated with HIV infection, including mild, asymptomatic, and HIV-related dementia, are known as HAND. A number of processes, including complex and multivariate pathogenesis, contribute to neuronal damage and cognitive impairment in HAND [53]. It is a frequent side effect of HIV infection that has a direct effect on mental health, resulting in decreased quality of life, social isolation, anxiety, and sadness. HAND can manifest as moderate cognitive impairments or severe manifestations, such as HIV-associated dementia, accompanied by neurological problems such motor dysfunction and poor verbal fluency [54,55].

### 9.3. The Prevalence and Impact of HAM/TSP in HIV/HTLV-1 Co-Infected Patients

The prevalence of HTLV-1 varies significantly based on geographical location, social demographics, and individual risk factors [56]. HTLV-1 is prevalent in specific regions, including southeastern Japan, the Caribbean, parts of Africa, the Middle East, Australia, and the Pacific Islands of Melanesia [57]. HTLV-1 is particularly prevalent in several South American and Caribbean countries, including Brazil, Colombia, French Guiana, Guyana, Haiti, Jamaica, and Peru [39]. HTLV-1 can cause ATL and HAM/TSP, while HIV is associated with AIDS [13]. Given their shared transmission routes, HIV-infected individuals are at a higher risk of also being infected with HTLV-1 [58,59].

Co-infection with HIV and HTLV-1 has been demonstrated to be associated with an elevated risk of neurological complications, including TSP, also known as HAM (Figure 3). This elevated risk has been observed in comparison to HIV infection alone and has been associated with conditions such as crusted scabies, strongyloidiasis, extrapulmonary tuberculosis, pneumonia, and esophageal candidiasis [60,61]. Another study claims that HIV/HTLV-1 co-infection is associated with a faster progression to AIDS and a shorter lifespan compared to HIV infection alone [7]. Patients suffering from co-infection with HTLV-1 and HIV have been observed to present with a number of common ailments, including asymmetric weakness in both lower limbs, lower back pain and urinary incontinence [62]. HAM/TSP is a neurological condition that causes progressive muscle weakness, particularly in the lower extremities. Additionally, it results in lower back discomfort and urinary difficulties. The disease progresses through two principal stages. Initially, there is an inflammatory phase during which the spinal cord becomes inflamed. Subsequently, there is a scarring and degeneration phase during which the nerves begin to deteriorate [63,64].

### 9.4. Management Strategies for HAM/TSP in the Context of ART

The initial identification of the association between HTLV-1 infection and TSP/HAM occurred in 1985 in Martinique, a French overseas department situated in the Caribbean region [65]. According to the WHO, it is estimated that between 5 and 10 million people worldwide are currently infected with HTLV-1. To date, there is no specific treatment or vaccine for HTLV-1 infection. Researchers found that ART, which was originally generated for treatment against HIV, is very effective against HTLV-1 due to the diseases’ genetic similarities [66]. ARTs are mainly focused on managing the symptoms of the disease rather than directly eliminating the HTLV-1 virus from the body, and researchers are exploring new drugs that could potentially reduce the amount of HTLV-1 in the body by influencing the immune system [33,34,67].

ART regimens typically include a combination of drugs from different classes to target HIV replication at multiple points in its life cycle [33]. The following classes are amongst the most commonly used: Nucleoside Reverse Transcriptase Inhibitors (NRTIs), Non-Nucleoside Reverse Transcriptase Inhibitors (NNRTIs), Integrase Strand Transfer Inhibitors (INSTIs) and Protease Inhibitors [33].

## 10. Public Health Implications of HIV/HTLV-1 Co-Infection

### 10.1. Epidemiology of Co-Infection in Different Geographic Regions

Research indicates that co-infection with HTLV-1/HIV-1 alters the normal course of HIV-1, accelerates the development of the disease to AIDS, and reduces survival duration. HIV-1 increases the expression of HTLV-1, which raises the risk of related illnesses including adult T-cell leukemia and TSP/HAM. Co-infection promotes HIV replication, raises CD4 levels, and indicates more severe HIV clinical illnesses [68]. Research has also identified intravenous drug use, multiple partner sexual contact, and prior blood transfusions as risk factors for HIV/HTLV co-infection [69,70,71].

In the 1980s, co-infections between HIV and HTLV-1 were reported in Europe, America, and Africa; the most prevalent populations were sex workers, hemophiliacs, drug users, and homosexuals. These days, most research on HTLV-1 and HIV-1 co-infection is focused on South America and Africa, where the prevalence ranges from 0.5 to 10.9%. Co-infection between HTLV-1 and HIV-1 is more commonly observed in highly endemic nations in South America, the Caribbean, and Africa; these countries are mostly in Latin America, sub-Saharan Africa, and Romania [34,68,72]. In Spain, less than 5% of HTLV-1 carriers co-occur with HIV. Patients that are also co-infected are predominantly from Latin America [72]. According to the study, HIV/HTLV co-infection is extremely uncommon in Bahia, Brazil; it is found in just 2.4% of all HIV-positive individuals and 3.4% of towns. Regions with greater rates of co-infection were hotspots for HIV and HTLV, which corresponded to significant commercial or tourism hubs [71].

Research conducted in African nations indicates that the incidence of co-infection between HIV-1 and HTLV-1 ranges from 1.55% to 3.9%. This underscores the significance of both vertical and sexual transmission of HTLV-1 as well as blood transfusion. For example, female prisoners in Mozambique had a co-infection prevalence of 3.45%, whereas in Guinea Bissau, West Africa, the rate was 2.8% [73,74].

### 10.2. Strategies for Public Health to Manage and Prevent Co-Infections

In order to reduce the risk of delayed therapy and death for co-infected patients, it is imperative to implement a test-and-treat strategy for HIV-positive patients living in areas endemic to HTLV infection. HIV-1 and HTLV-1 co-infection as well as HIV-1 and HTLV-1/2 triple infection are linked to lower survival rates, death, and faster progression to death [10]. Because of inadequate understanding of HTLV/HIV co-infection, research on epidemiology, pathophysiology, and treatment outcomes must be conducted internationally. HIV antiretrovirals have poor efficacy in treating HTLV-1 because of their molecularly tailored character, and there are currently no conventional therapies available [68,75].

Studies show that individuals who co-infect with HIV-1 and HTLV-1/2 require preventative and management strategies. For possible clinical ramifications, elevated CD4^+^ cell counts should be investigated for HTLV-I co-infection. To prevent co-infected patients from becoming worse, medication should be guided by quantitative virologic data [10].

### 10.3. Diagnostic Challenges in Identifying and Managing HIV/HTLV-1 Co-Infection

Given the potential for HTLV-I to accelerate HIV disease progression, the interplay between these two viruses has been extensively investigated within the scientific community [76]. The clinical management of HIV-1/HTLV-1 co-infection presents a significant challenge [77]. Patients with co-infections of HIV/HTLV-1 typically exhibit significantly elevated CD4^+^ T-cell counts in comparison to those with HIV-1 mono-infection [78]. Consequently, the diagnosis of AIDS using these criteria is rendered unreliable, resulting in a shortened lifespan for the patients [78]. HIV-1 has been observed to reduce the number of lymphocytic cells and exert a cytopathic effect, whereas HTLV-1 has been demonstrated to stimulate the proliferation of CD4^+^ T-cells without exhibiting cytopathic effects.

It is possible that co-infection may exacerbate the progression of AIDS and promote the occurrence of future infections, due to the fact that HIV-1’s immunosuppression may be masked. The risk of developing TSP or HAM is increased in individuals infected with both HIV-1 and HTLV-1 [79]. Individuals infected with both HIV and HTLV-1 who are undergoing ART are at an elevated risk of developing neurological complications over the course of their lifetime. The administration of zidovudine, lamivudine, and abacavir to treat HIV-1 infection in patients who were also HIV-1/HTLV-1 co-infected resulted in an increase in the proviral load of HTLV-1 [40].

### 10.4. Barriers to Effective Treatment and Management, Especially in Resource-Limited Settings

ART has revolutionized HIV care by significantly mitigating morbidity and mortality. However, achieving optimal clinical outcomes necessitates consistent adherence to treatment regimens and equitable access to ART. Over 90% of HIV-positive individuals live in resource-limited settings, where statistics on barriers to care are beginning to emerge. These obstacles encompass a spectrum of challenges, including low rates of HIV testing, difficulties in connecting individuals to HIV care services, and suboptimal medication adherence [80]. To gain access to ART, a patient must successfully complete a series of care phases, including diagnosis, linkage, disease state monitoring and treatment initiation [81]. Mental health, particularly depression, has been identified as a significant factor influencing the quality of HIV care received by people living with HIV in South Africa. A study conducted in KwaZulu-Natal revealed that individuals with depression were less likely to initiate or maintain ART. The implementation of routine mental health assessments during the course of HIV testing may facilitate the identification of individuals with mental health disorders and enhance the overall efficacy of HIV treatment [82].

## 11. Perspectives and Future Challenges

While the studies discussed above have illuminated aspects of the immune development system in HIV and HTLV co-infections, the mechanisms that regulate these interactions remain elusive. HIV and HTLV-1/2 co-infections may be more common than currently recognized, as routine testing for HTLV-1/2 in outpatient HIV clinics is not typically recommended. Co-infection with HTLV-1 and HTLV-2 appears to have distinct effects on individuals with HIV. Specifically, HTLV-1 may accelerate clinical progression to AIDS, while HIV infection may increase the risk of HTLV-1-associated diseases.

Recently, the Tax and Rex viral proteins of HTLV-1 have garnered considerable attention in clinical medicine. Research has demonstrated that Tax oncoproteins disrupt interferon (IFN) production and signaling in cells infected with HTLV-1/2. A deeper understanding of the molecular mechanisms underlying this disruption could facilitate the development of strategies to counteract Tax and restore the innate IFN response against viral infections. For instance, agonists of both RIG-I and STING have been identified to possess broad-spectrum antiviral properties [83]. However, further studies are necessary to elucidate the precise functions of these proteins.

HTLV-1/2 infections are chronic and currently lack effective treatments or cures. To curb the spread of HTLV-1, the development of a safe and effective vaccine is essential. Although the HBZ protein is a potential target for ATL treatment, the efficacy of vaccines based on recombinant viruses expressing HBZ or HBZ peptides requires further evaluation in human ATL cases [84]. A significant challenge with antigenic peptides is their low immunogenicity [85]. HTLV-1 and HTLV-2 co-infections have varying impacts on individuals with HIV. HTLV-1 may accelerate the progression to AIDS and increase the risk of HTLV-1 associated diseases, although some data are contradictory. Conversely, HTLV-2 co-infection appears to have a protective effect, reducing the progression to AIDS. A common factor in HIV patients co-infected with either HTLV-1 or HTLV-2 is elevated CD4 counts. There is definitely a need for larger, well-designed studies to clearly determine the impact of HIV/HTLV-1/2 co-infection.

## 12. Conclusions

In the absence of therapeutic strategies or standardized patient care for individuals co-infected with HIV and HTLV-1/HTLV-2, there is an urgent need for the development of antiviral treatments and vaccines. The genetically conserved 5′ and 3′ LTR regions of HTLV-1 are promising targets for epigenetic silencing. Despite the innovative potential of RNA interference (siRNA) therapeutics, significant challenges remain, including the efficient and targeted delivery of these therapies and their clinical translation. Addressing these challenges is crucial for advancing the field and improving patient outcomes.

This review has emphasized the immunological and neurological perspectives of HIV-1 and HTLV-1. It also explored recent progress in elucidating the molecular determinants of HTLV-1 and HIV-1 assembly, with a particular focus on the interactions between the Tax protein and other associated gene functions. Despite these advancements, critical aspects of HTLV-1 replication, including virus entry, uncoating, reverse transcription, assembly, and budding, remain inadequately understood.

## Figures and Tables

**Figure 1 viruses-17-00545-f001:**
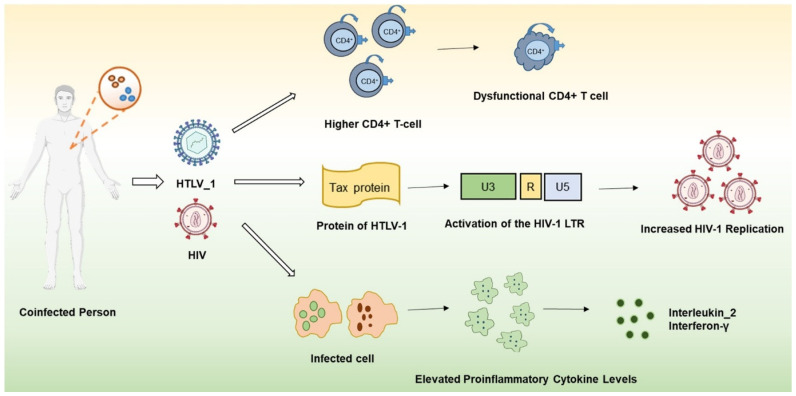
**Dysregulation of CD4^+^ T cells and inflammation in HIV and HTLV-1 co-infected patients.** The presence of both HIV and HTLV-1 in an individual’s immune system gives rise to a multifaceted immune dysregulation, marked by a paradoxical expansion of CD4^+^ T cells, the presence of chronic inflammation, and an escalated disease burden. Despite elevated CD4^+^ counts, immune function is impaired due to ongoing activation and exhaustion. During HTLV transcription, the Tax protein is upregulated in the presence of HIV through the action of nuclear factor kappa B (NF-κB). This upregulation mechanism is similar to that of cytokines such as interleukin. Within the nucleus, Tax assembles and modifies subcellular structures called Tax nuclear bodies (Tax NBs), where it enhances NF-κB transcriptional activity to its maximum potential. Graphics created by BioRender.com.

**Figure 2 viruses-17-00545-f002:**
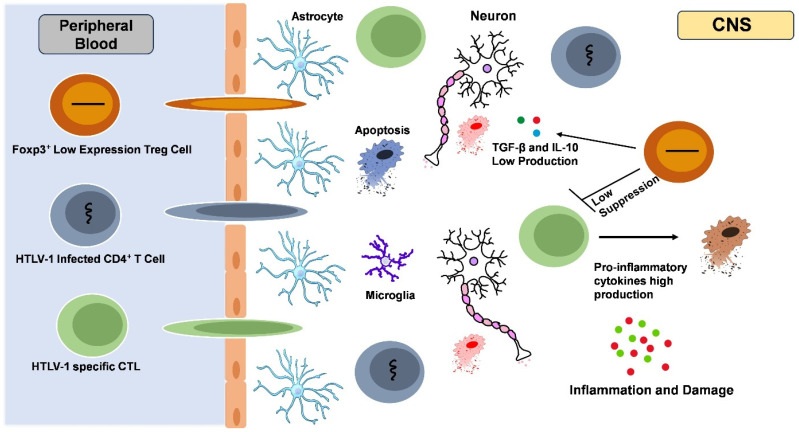
**Neurological impact of HIV and HTLV-1 co-infection.** The impact of HIV and HTLV-1 co-infections on the nervous system is significant, manifesting as a complex interplay of neuroinflammation, immune dysregulation, and neurodegeneration. While HIV is known to cause HAND due to chronic inflammation and direct neuronal toxicity, HTLV-1 is linked to HTLV-1-associated myelopathy/tropical spastic paraparesis (HAM/TSP), a progressive neuroinflammatory disorder affecting the spinal cord. HAM/TSP is the most common neurological manifestation of HTLV-1 and HIV co-infections. Astrocytes are the most abundant cell type in the brain and the only source of HIV and HTLV-1. HIV from infected astrocytes egresses, likely through trafficking of CD4^+^ T cells, into peripheral organs, as indicated by the detection of HIV DNA/RNA. Graphics created by BioRender.com.

**Figure 3 viruses-17-00545-f003:**
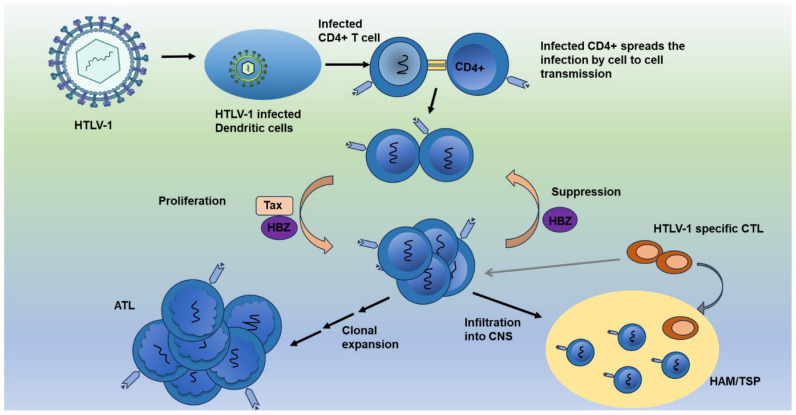
**Exploring the pathogenesis of HAM/TSP and its impact on HTLV-1-induced neurological effects.** This figure explores the pathogenesis of HAM/TSP and highlights the neurological effects induced by HTLV-1 infection, with a particular focus on individuals co-infected with HIV. HTLV-1 is associated with a range of immune-mediated disorders and lymphoproliferative malignancies, with ATL and HAM/TSP being the most prominent consequences. These conditions are the result of a highly selective process, occurring in only a small fraction of individuals infected with HTLV-1 or co-infected with both HTLV-1 and HIV. HTLV-1 causes lymphoproliferative malignancies and many immune-mediated disorders. ATL and HAM/TSP are the consequence of a highly selective process that occurs in only a fraction of infected individuals or those co-infected with HTLV and HIV. The figure further illustrates the role of the HTLV-1 Tax protein, which plays a key role in the pathogenesis of these disorders. The Tax protein activates HIV replication by interacting with the HIV-1 Long Terminal Repeat (LTR), which can lead to enhanced viral replication in co-infected individuals. Graphics created by BioRender.com.

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
