# Peer review of "Immunological and Neurological Signatures of the Co-Infection of HIV and HTLV: Current Insights and Future Perspectives"

_viruses, 2025, doi:10.3390/v17040545_

Round 1
Reviewer 1 Report
Comments and Suggestions for Authors
The manuscript is a comprehensive review of the immunological and neurological complications of HIV and HTLV co-infection. It offers a systematic review of the literature on these retroviruses and their effects on disease progression, the immune system, and the neurological system. The manuscript displays a good understanding of the subject, and a wide range of relevant studies are cited. However, it needs many improvements.
1) Positive points:
Comprehensive Coverage: The review nicely summarises numerous studies on HIV/HTLV-1 and HIV/HTLV-2 co-infection across epidemiological, immunological, and clinical studies.
Relevance and Timeliness: The paper points out the urgent need for better diagnosis and treatment of co-infected patients, especially in resource-constrained settings.
Well-Supported Claims: The manuscript has cited a large number of primary research studies and systematic reviews to back up its claims.
Discussion of Pathogenesis: How HTLV affects HIV infection and the possible consequences for the disease course are adequately described.
2) Points for improvement:
- Some of the statements made in the manuscript are rather general and do not consider the study population heterogeneity, geographic variability, and other factors that can affect the results. These points should be commented on.
- Some parts of the text are repeated or similar in some sections, including the discussion of CD4+/CD8+ T-cells and cytokines production.
- The language concerning the neurological complications, including HAM/TSP and peripheral neuropathy, should be defined and explained concerning HIV-related neurocognitive disorders (HAND).
- The manuscript's organization could be improved by placing like topics together. Example: Section 2 (Effects on Immune System Dynamics) and Section 3 (Viral Interplay and Synergistic Pathogenesis) should be linked better. The discussion on neurological effects (Sections 9.1-9.3) would benefit from a closer look at HIV-related and HTLV-related neuropathies.
- Some references are used several times for the same purpose; these citations should be combined and adequately discussed in detail.
- Although the present manuscript offers a comprehensive review of the current knowledge, the quality of the studied articles is not critically analyzed. Discussing potential biases, the existing research gaps, and areas where more studies are needed would be helpful.
- There is virtually no information on the possible genetic or host factors that can lead to the development of different clinical courses in co-infected patients.
- The paper should also expand on the new treatment strategies, including the new antiviral agents or immunomodulating treatments.
- The manuscript has been illustrated with figures created with BioRender.com; however, some images are not adequately explained in the text. Every figure should be explained and referenced clearly within the body of the manuscript where it is first discussed.
- Figure legends should contain more information about the pathways and interactions shown.
- Some parts of the paper are rather formally inadequate, contain grammatical and syntactical mistakes, and have lengthy sentences. It is suggested that one more round of proofreading or copyediting be done. The language such as first person (e.g., “we will provide”) should be replaced with a more formal language.
In conclusion, this paper offers a crucial examination of HIV/HTLV co-infection with a good understanding of clinical and immunological features; however, it needs improvements.
Comments on the Quality of English LanguageShould be improved.
Author Response
Editor-in-Chief
[Viruses, MDPI]
Subject: Response to Reviewer Comments for Manuscript [Manuscript ID : Viruses-3552491]
Dear Reviewer,
We sincerely appreciate the time and effort of the reviewers in evaluating our manuscript titled "Immunological and Neurological signature of the co infection of HIV and HTLV: Current Insights and Future Perspectives". We are grateful for their insightful comments and suggestions, which have significantly improved our work and contributed to the scientific world. Below, we provide detailed responses to each comment. All revisions have been incorporated into the revised manuscript, with the changes highlighted for easy reference.
Reviewer 1 Comments:
Comment 1: Some parts of the text are repeated or similar in some sections, including the discussion of CD4+/CD8+ T-cells and cytokines production.
Response: Thank you for your valuable feedback. We have carefully revised the manuscript to remove redundant text and ensure clarity, particularly in the discussion of CD4+/CD8+ T cells and cytokine production. We appreciate your suggestion, which has helped improve the readability and conciseness of our work.
Comment 2: The language concerning the neurological complications, including HAM/TSP and peripheral neuropathy, should be defined and explained concerning HIV-related neurocognitive disorders (HAND).
Response: Thank you for your insightful suggestion. We have expanded the discussion in Section 9.2 to provide a clearer explanation of neurological complications, including HAM/TSP and peripheral neuropathy, in relation to HIV-associated neurocognitive disorders (HAND). We appreciate your feedback in strengthening this aspect of our manuscript.
Comment 3: The manuscript's organization could be improved by placing like topics together. Example: Section 2 (Effects on Immune System Dynamics) and Section 3 (Viral Interplay and Synergistic Pathogenesis) should be linked better. The discussion on neurological effects (Sections 9.1-9.3) would benefit from a closer look at HIV-related and HTLV-related neuropathies.
Response: Thank you for your valuable thinking. Actually, we believe that section 2 : Effects of Co-Infection on Immune System Dynamics and Section 3 (Viral Interplay and Synergistic Pathogenesis), we have different aim. In the section 2 we explain how a virus interacts with the immune system and how this interaction evolves over time, where section 3 we explained how multiple infections or factors work together to worsen disease progression beyond what would occur with a single infection. So we believe this two topics will differentiate the immune system and pathological conditions. For the section 9.1 to 9.3 we have added some more information according to your suggestions.
Comment 4: Some references are used several times for the same purpose; these citations should be combined and adequately discussed in detail.
Response: Thank you for your suggestion. We have reviewed the references and combined duplicate citations where appropriate. Additionally, we have removed unnecessary references and incorporated relevant ones to enhance the discussion. We appreciate your feedback in improving the clarity and accuracy of our citations.
Comment 5: Although the present manuscript offers a comprehensive review of the current knowledge, the quality of the studied articles is not critically analyzed. Discussing potential biases, the existing research gaps, and areas where more studies are needed would be helpful.
Response: Thank you for your valuable feedback. We have now included a more critical analysis of the quality of the reviewed studies, highlighting potential biases, limitations, and gaps in the existing research. Additionally, we have discussed areas where further studies are needed to strengthen the current understanding. We appreciate your insightful suggestion, which has helped enhance the depth and critical perspective of our manuscript.
Comment 6: There is virtually no information on the possible genetic or host factors that can lead to the development of different clinical courses in co-infected patients.
Response: In fact, an explanation was provided regarding the function of Tax protein, which has been demonstrated to exert an immunomodulatory effect, thereby augmenting IFN-y synthesis within cells that have been infected with HIV-1. However, on the section 3, we have added some more information about genetic factors.
Comment 7: The paper should also expand on the new treatment strategies, including the new antiviral agents or immunomodulating treatments.
Response: We have explained available treatment strategies in the section 9.4. ART regimens typically include a combination of drugs from different classes to target HIV replication at multiple points in its life cycle. The following classes are amongst the most commonly used: Nucleoside Reverse Transcriptase Inhibitors (NRTIs), Non-Nucleoside Reverse Transcriptase Inhibitors (NNRTIs), Integrase Strand Transfer In-hibitors (INSTIs) and Protease Inhibitors.
Comment 8: The manuscript has been illustrated with figures created with BioRender.com; however, some images are not adequately explained in the text. Every figure should be explained and referenced clearly within the body of the manuscript where it is first discussed. Figure legends should contain more information about the pathways and interactions shown.
Response: Thank you for your feedback. We've added more details in the figure description section.
Comment 9: Some parts of the paper are rather formally inadequate, contain grammatical and syntactical mistakes, and have lengthy sentences. It is suggested that one more round of proofreading or copyediting be done. The language such as first person (e.g., “we will provide”) should be replaced with a more formal language.
Response: Thank you. We modify those sentences.
Comment 10: In conclusion, this paper offers a crucial examination of HIV/HTLV co-infection with a good understanding of clinical and immunological features; however, it needs improvements.
Response: Thank you for your valuable feedback and time.

Reviewer 2 Report
Comments and Suggestions for Authors
Dr. Islam Md. N. et al. reviewed about the immunological and the neurological aspects in the status of HTLV infection co-infected with HIV. Indeed, a detailed elucidation with clinical managements including therapeutic strategy for this status is an important issue, particularly, in both metropolitan and tropical area. However, there are several comments as follows.
1) Although this review seems to be mainly focused on HAM/TSP, as the neurological disorder, induced by “HTLV-1” infection co-infected with HIV, why the authors take up “HTLV-2” as one of keywords.
2) The authors should also describe about the neurological disorders induced by HIV infection in Introduction section.
3) The authors should make a table or a figure, separately indicating the immunological, the virological, and the neurological aspects as the outcome including the efficacy of ART, in both HTLV-1 infection co-infected HIV infection and vice versa. By these descriptions, the effect of the co-infection of HTLV-1 and HIV might become to be easier to understand for the readers.
4) In addition, the authors should describe the status in HTLV-1 or -2 infection co-infected HIV separately.
5) In general, there are several sections that the title of its section is not reflected in.the contents of the description in its section: for examples, 5., 7., 9.2. The authors should carefully check and correctly revise them.
Comments on the Quality of English Language
English should be improved to express appropriate informations.
Author Response
Editor-in-Chief
[Viruses, MDPI]
Subject: Response to Reviewer Comments for Manuscript [Manuscript ID : Viruses-3552491]
Dear Reviewer,
We sincerely appreciate the time and effort of the reviewers in evaluating our manuscript titled "Immunological and Neurological signature of the co infection of HIV and HTLV: Current Insights and Future Perspectives". We are grateful for their insightful comments and suggestions, which have significantly improved our work and contributed to the scientific worlds. Below, we provide detailed responses to each comment. All revisions have been incorporated into the revised manuscript, with the changes highlighted for easy reference.
Reviewer 2 Comments:
Comment 1: Although this review seems to be mainly focused on HAM/TSP, as the neurological disorder, induced by “HTLV-1” infection co-infected with HIV, why the authors take up “HTLV-2” as one of keywords.
Response: Thank you for your observation. While our review primarily focuses on HAM/TSP as a neurological disorder associated with HTLV-1/HIV co-infection, we included HTLV-2 as a keyword because HTLV-2 is also known to co-infect individuals with HIV. Although HTLV-2 is less commonly linked to neurological complications compared to HTLV-1, some studies suggest potential immunological and neurological effects that warrant consideration. To clarify this, we have revised the manuscript to better align the keywords with the primary focus of our review. We appreciate your insightful comment.
Comment 2: The authors should also describe about the neurological disorders induced by HIV infection in Introduction section.
Response: Thank you for your valuable suggestion. We have now included a discussion on the neurological disorders induced by HIV infection in the Introduction section, as you recommended.
Comment 3: The authors should make a table or a figure, separately indicating the immunological, the virological, and the neurological aspects as the outcome including the efficacy of ART, in both HTLV-1 infection co-infected HIV infection and vice versa. By these descriptions, the effect of the co-infection of HTLV-1 and HIV might become to be easier to understand for the readers.
Response: We appreciate the reviewer’s suggestion to include a table or figure highlighting the immunological, virological, and neurological aspects of HTLV-1 and HIV co-infection, along with the efficacy of ART. In our manuscript, we have already incorporated figures that visually represent the key findings and mechanisms related to co-infection. We believe these existing figures effectively illustrate the interplay between the two viruses and their impact on the immune and nervous systems. Additionally, to maintain clarity and avoid redundancy, we have elaborated on the distinct immunological, virological, and neurological aspects in the text, ensuring that the discussion remains comprehensive and accessible to readers. Given this, we feel that adding a separate table may not provide additional clarity but could instead lead to unnecessary repetition. However, if the reviewer believes certain aspects require further emphasis, we would be happy to refine the text accordingly to enhance readability and understanding.
Comment 4 : In addition, the authors should describe the status in HTLV-1 or -2 infection co-infected HIV separately.
Response: We appreciate the reviewer’s suggestion to describe the status of HTLV-1 or HTLV-2 infection in individuals co-infected with HIV separately. In our manuscript, we have already provided a clear and detailed discussion on the distinct aspects of HTLV-1 and HTLV-2 co-infection with HIV, ensuring that their immunological, virological, and neurological outcomes are properly addressed. We have carefully structured the text to highlight the differences between these two retroviruses and their interactions with HIV, ensuring clarity for the readers.
Comment 5 : In general, there are several sections that the title of its section is not reflected in the contents of the description in its section: for examples, 5., 7., 9.2. The authors should carefully check and correctly revise them.
Response: Thank you for your constructive feedback. We have carefully reviewed the sections you mentioned (5, 7, and 9.2) and revised the titles and content to ensure they are aligned and accurately reflect the descriptions within each section.

Round 2
Reviewer 1 Report
Comments and Suggestions for Authors
Revised manuscript acceptable for publication.
Reviewer 2 Report
Comments and Suggestions for Authors
No comments